

# Research and progress on the mechanism of lower urinary tract neuromodulation: a literature review

Shutong Pang[1] and Junan Yan[1,2]

[1] Guangxi Key Laboratory of Special Biomedicine and Advanced Institute for Brain and Intelligence, School of Medicine, Guangxi University, Nanning, Guangxi, China
[2] Department of Urology, PLA Naval Medical Center, Naval Medical University, Shanghai, China

## ABSTRACT

The storage and periodic voiding of urine in the lower urinary tract are regulated by a complex neural control system that includes the brain, spinal cord, and peripheral autonomic ganglia. Investigating the neuromodulation mechanisms of the lower urinary tract helps to deepen our understanding of urine storage and voiding processes, reveal the mechanisms underlying lower urinary tract dysfunction, and provide new strategies and insights for the treatment and management of related diseases. However, the current understanding of the neuromodulation mechanisms of the lower urinary tract is still limited, and further research methods are needed to elucidate its mechanisms and potential pathological mechanisms. This article provides an overview of the research progress in the functional study of the lower urinary tract system, as well as the key neural regulatory mechanisms during the micturition process. In addition, the commonly used research methods for studying the regulatory mechanisms of the lower urinary tract and the methods for evaluating lower urinary tract function in rodents are discussed. Finally, the latest advances and prospects of artificial intelligence in the research of neuromodulation mechanisms of the lower urinary tract are discussed. This includes the potential roles of machine learning in the diagnosis of lower urinary tract diseases and intelligent-assisted surgical systems, as well as the application of data mining and pattern recognition techniques in advancing lower urinary tract research. Our aim is to provide researchers with novel strategies and insights for the treatment and management of lower urinary tract dysfunction by conducting in-depth research and gaining a comprehensive understanding of the latest advancements in the neural regulation mechanisms of the lower urinary tract.

Corresponding author
Junan Yan, pest1976@qq.com

## INTRODUCTION

The lower urinary tract (LUT) relies on the coordinated activity of the urinary reservoir (bladder) and the outlet composed of the bladder neck, urethra, and urethral sphincters for the storage and periodic elimination of urine. Urinary storage and excretion constitute an extremely complex physiological process that is controlled by the brain, spinal cord, and peripheral nerves, involving various neurotransmitters (*De Groat, 1998*; *Karnup & De*

*Groat, 2020*; *Yoshimura & De Groat, 1997*). To gain a comprehensive understanding of the neuromodulation mechanisms of the LUT, researchers have employed various methods such as anatomy, neurophysiology, and optogenetic in different animal models. These methods enable accurate labeling and tracking of neuronal connections and activities (*Fry et al., 2010*; *Hutch, 1967*). Furthermore, noninvasive tests such as the void-spot assay and the metabolic cage , as well as more invasive urodynamics investigations, are applied to evaluate the LUT function in animals (*Sartori, Kessler & Schwab, 2021*). A recent study was the first to compare voiding parameters assessed by metabolic cage and fully awake urodynamic models, demonstrating that catheter implantation does not significantly affect physiological bladder function. This also indicates that the urodynamic model in awake animals is currently the most suitable translational method, as it allows for repetitive analysis at different time points in the same animal, thereby opening promising avenues for the investigation of LUT function (*Schneider et al., 2018*).

However, the neural regulation of the LUT involves complex interactions between multiple organs and neurons. Despite significant advancements in understanding the intricate neural control of the seemingly simple process of urination in recent years, traditional research and evaluation methods still struggle to fully capture and interpret this complexity (*Fowler, Griffiths & De Groat, 2008*). In recent years, the rapid development of artificial intelligence has provided new insights into the study of neural regulation mechanisms in the LUT. For instance, the application of machine learning and deep learning can aid in deciphering the complex structure and functions of neural networks, thereby revealing the underlying mechanisms of the nervous system (*Neghab et al., 2022*). Moreover, artificial intelligence techniques can be employed to establish simulation and virtual experimental platforms for studying neural regulation of the LUT. These platforms can simulate the physiological processes and disease mechanisms of urinary regulation, assisting researchers in theoretical exploration and experimental validation, thus expediting the development of novel treatment methods and technologies (*Li et al., 2021*).

Therefore, this review article summarizes the following aspects: (1) the functional studies of the LUT system and the neural regulation mechanisms of urinary voiding; (2) the research methods currently employed in investigating LUT neural regulation mechanisms; (3) the current methods for assessing LUT function; and (4) the applications and prospects of artificial intelligence in the study of neural regulation in the LUT. By gaining an in-depth understanding of the research and advancements in the neural modulation mechanisms of the LUT, we hope to provide new strategies and insights for the treatment and management of LUT dysfunction.

## SURVEY METHODOLOGY

We conducted a systematic literature search using the Web of Science database and PubMed database, based on the following keywords: (1) Lower urinary tract anatomy, (2) lower urinary tract function, (3) urology neural control, (4) urology and electrophysiological recording, (5) urology and optogenetics, (6) urodynamics, (7) void spot assay, (8) urology and artificial intelligence. We excluded studies that were not relevant to the neural

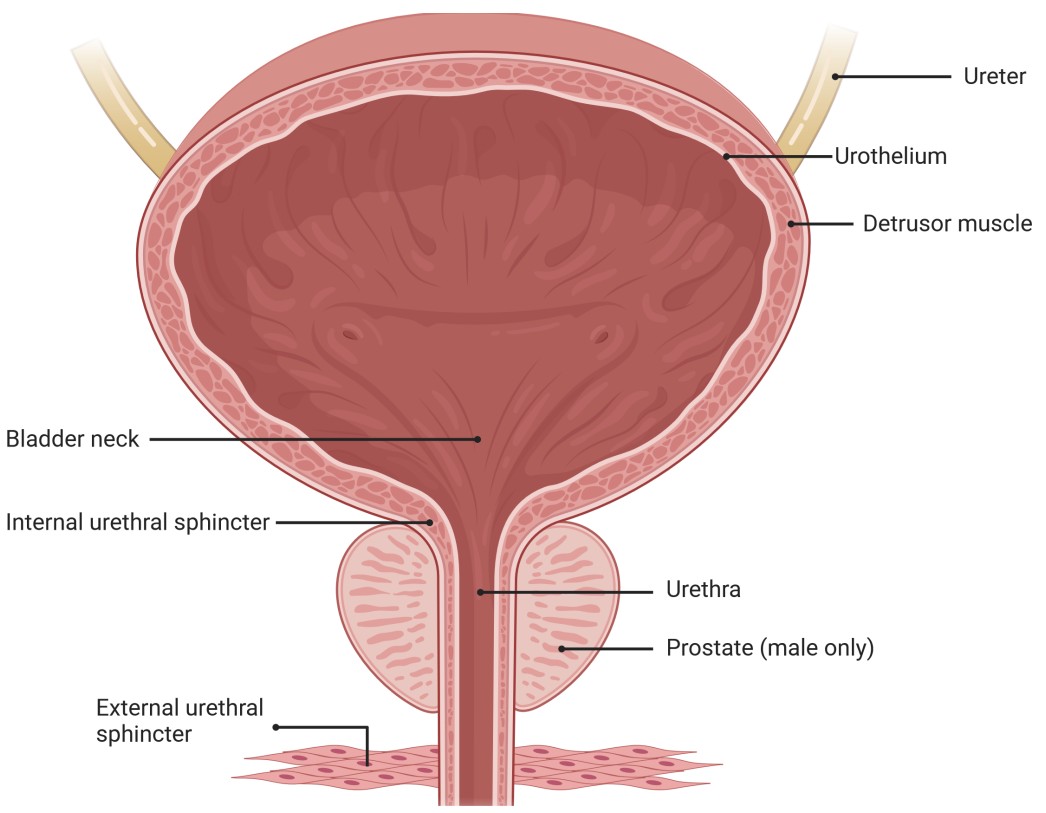

**Figure 1** **Anatomical structure of the lower urinary tract.** Urinary storage and periodic urination in the LUT depend on the coordinated activity of the bladder, urethra, and the striated muscles of the urethral sphincters. Image components credit: https://www.biorender.com/features.

regulation mechanisms of the lower urinary tract in humans or animals. Our search criteria did not involve further refinements based on publication dates, journal impact factors, authors, or author affiliations, as per the SCI standard.

# THE NEURAL REGULATORY MECHANISMS OF LUT FUNCTION

## Functional studies of the LUT

The LUT is a crucial component of the mammalian urination system, relying on the coordinated function of two functional units: the bladder and the outlet composed of the urethra and the striated muscle of the urethral sphincter (Fig. 1). These units work together to ensure the storage and effective expulsion of urine (*De Groat, 1998*; *Sartori, Kessler & Schwab, 2021*; *Yao et al., 2018*; *Yoshimura & De Groat, 1997*).

The bladder is a sac-like organ located at the bottom of the pelvic cavity, responsible for storing continuously secreted urine and automatically expelling it at appropriate times. Unlike the phasic contractions observed in other visceral organs, the bladder controls both reflexive and voluntary urination through a switch-like mechanism (*Birder, 2013*; *DeSesso, 1995*; *Hickling, Sun & Wu, 2015*; *Hicks, 1975*). Traditionally, it was believed that sensory

signals in the bladder were primarily mediated by direct activation of afferent nerves within the bladder. However, recent research has revealed the influence of non-neuronal cells, such as urothelial cells, on the sensory system. These non-neuronal cells are capable of responding to various stimuli, including physiological, psychological, and disease-related factors (*Birder, 2013*). Recent studies have demonstrated, using optogenetics techniques in mouse models, that direct stimulation of urothelial cells alone is sufficient to induce bladder contractions (*Robilotto et al., 2023*). Furthermore, neurotransmitters including acetylcholine (ACh), adenosine triphosphate (ATP), adenosine, nitric oxide (NO), and prostaglandins released by urothelial cells play a crucial role in the precise regulation of the sensory network in the bladder wall. These neurotransmitters not only contribute to bladder sensory perception and normal micturition but also have implications in various bladder disorders. However, current research on the specific mechanisms by which non-neuronal cells affect bladder function still lacks sufficient and conclusive data, necessitating further in-depth investigations (*Grundy et al., 2023*; *Winder et al., 2014*). Due to the complexity of the human bladder and interspecies differences, an ideal model that fully simulates human bladder function has yet to be developed. Therefore, the selection of suitable animal models is essential for research purposes, including rodents, rabbits, dogs, pigs, and cats (*Shen et al., 2021*).

The urethra, as a conduit connected to the bladder neck, serves to connect the bladder with the external environment for urine passage. At the proximal end of the urethra, muscle fibers form the urethral sphincter, which controls the expulsion of urine from the bladder through the urethra (*Hill, 2015*; *McNeal, 1972*; *Pradidarcheep et al., 2011*). Traditionally, the urethra has been primarily regarded as a conduit for guiding urine from the bladder to the outside (*Heesakkers & Gerretsen, 2004*; *Wei & De Lancey, 2004*; *Yao, Hou & Yin, 2012*; *Yucel & Baskin, 2004*). However, accumulating evidence suggests that the urethra may have a more significant role in local regulation than previously recognized and may directly influence bladder function through LUT mechanisms. Recent studies have indicated the existence of a urethra-bladder reflex that promotes voiding, a process that may depend on urethral sensory nerve innervation. In fact, evidence suggests the presence of specific subpopulations of submucosal neurons in the urethral region expressing neurotransmitters such as acetylcholine and serotonin (*Coelho et al., 2018*; *Deckmann & Kummer, 2016*). These cells are located in proximity to nerve fibers within the lamina propria and are capable of releasing neurotransmitters to rapidly induce detrusor muscle contractions, supporting the existence of urethra-bladder crosstalk. These neurotransmitters may activate urethral sensory fibers and trigger reflexive contractions of the bladder. However, further research is needed to fully understand the mechanisms of urethral sensation and gain a better understanding of the regulation of urethral sensory processes. Such investigations may contribute to the discovery of new therapeutic targets for the treatment of LUT dysfunctions, including urinary incontinence, detrusor-sphincter dyssynergia, and stress urinary incontinence (*Ferreira & Duarte Cruz, 2021*; *Pipitone, Sadeghi & De Lancey, 2021*).

In conclusion, the LUT plays a significant role in the urinary system. Understanding their structure and function is crucial for diagnosing and treating issues related to urine
storage and excretion. Additionally, researchers need to continue their efforts in developing better models and treatment methods to address diseases and problems associated with the LUT.

## The neural regulatory mechanisms of the LUT

The autonomic control of the LUT involves complex interactions, including autonomic nerves (mediated by sympathetic and parasympathetic nerves) and somatic nerves (mediated by the pudendal nerve) efferent pathways (*Fowler, Griffiths & De Groat, 2008*). Neural circuits extending from the cerebral cortex to the bladder maintain the regulation of micturition, permitting urination in appropriate social contexts (*Tish & Geerling, 2020*).

During bladder filling, the sympathetic nervous system primarily controls the process. Sympathetic nerves signal from the thoracolumbar spinal cord (T10-L2), releasing norepinephrine to act on $\beta$-adrenergic receptors in the detrusor muscle, causing its relaxation and enabling bladder filling. Simultaneously, sympathetic nerves act on $\beta$-adrenergic receptors in the smooth muscle of the urethral sphincter, causing contraction to prevent urinary leakage (*Danziger & Grill, 2016*; *Griffiths, 2015*; *Hill, 2015*; *Ito et al., 2020*; *McGuire, 1986*; *Merrill et al., 2016*). Sensory nerve endings in the bladder transmit filling signals to the spinal cord. These signals travel through the pelvic nerves to the sacral spinal cord and then ascend *via* the spinopontine pathways to the periaqueductal gray (PAG) in the midbrain. Neurons in the PAG further transmit signals to the insular cortex, registering the sensation of normal voiding desire. Subsequently, signals are relayed to the prefrontal cortex (PFC) to assess the safety and appropriateness of urination (Fig. 2A) (*Arya & Weissbart, 2017*; *Fowler, Griffiths & De Groat, 2008*).

In normal conditions, the PFC suppresses awareness of bladder filling by inhibiting the PAG. However, as bladder filling increases, signals sent to the insular cortex lead to conscious desire to urinate. This process activates the lateral prefrontal cortex to assess the safety and appropriateness of urination. If evaluation indicates inappropriate urination, inputs from the PFC suppress the pontine micturition center (PMC) *via* the anterior thalamic radiations to the brainstem. A group of neurons in the PMC expressing corticotropin-releasing hormone (CRH) and estrogen receptor 1 (ESR1) has recently been identified as critical for driving context-dependent micturition (*Hou, 2017*; *Keller et al., 2018*). Furthermore, recent studies have identified a small group of layer 5 neurons in the primary motor cortex (M1) that project to the PMC to modulate urination control (*Yao et al., 2018*). The inhibitory effect of the PMC influences the medial and lateral cell groups of the S2–S4 spinal segments, leading to reduced parasympathetic output and relaxation of the detrusor muscle. This relaxation state allows the bladder to continue storing urine without unnecessary emptying. However, when bladder distension reaches a critical level, afferent activity from tension receptors activates pathways to maximum activity state. Simultaneously, increased sympathetic discharge causes relaxation of the detrusor muscle and contraction of the urethral sphincter, promoting urine storage. This coordinated action enables the bladder to maintain urinary retention until an appropriate time for voiding (*Karnup, 2021*; *Unger et al., 2014*).

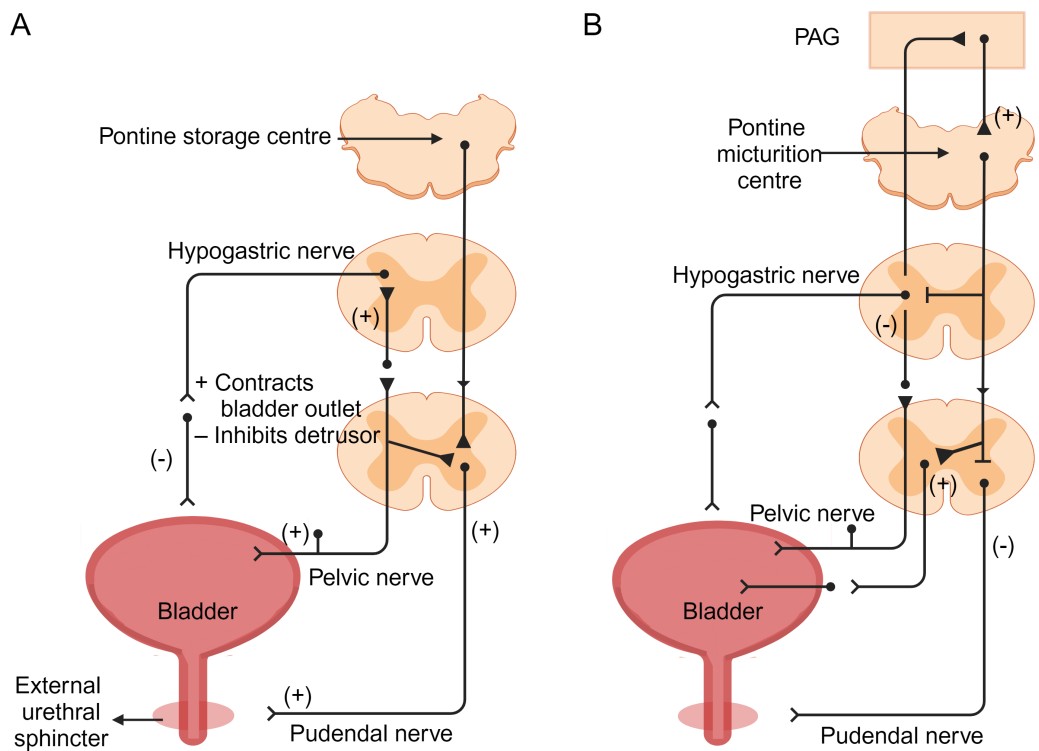

**Figure 2  Neural control of bladder filling and voiding (*Griffiths, 2015*; *Hill, 2015*).** (A) During the filling phase, bladder distension generates low-level bladder afferent discharge, which in turn stimulates sympathetic nerves to flow out from the sympathetic nerves of the inferior mesenteric ganglion to the bladder neck and bladder wall. This sympathetic stimulation relaxes the detrusor muscle and causes contraction of the internal sphincter at the bladder neck. Afferent impulses from the pelvic nerves also stimulate the external sphincter (somatic) to contract, resulting in contraction and maintenance of incontinence. (B) During the micturition process, there is high-intensity afferent activity signaling bladder distension, which activates the pontine micturition center in the brainstem. Strong bladder afferent discharge from the pelvic nerves activates the spinal cord-to-pontine micturition reflex pathway traversing the brainstem. This stimulation inhibits sympathetic outflow to the bladder and urethral smooth muscles, while activating parasympathetic outflow. Additionally, it inhibits somatic outflow to the urethral outlet. Ascending spinal cord input signals may relay through interneurons in the periaqueductal gray before reaching the pontine micturition center. Image components credit: https://www.biorender.com/features.

If the evaluation indicates that voiding is appropriate, the inhibitory signal from the higher centers of the PMC is released through the PAG. The neural signals from the PMC are transmitted *via* the sacral nerves to the pelvic nerve cells and then along the pelvic nerves to the bladder wall.These signals activate parasympathetic neurons in the medial and lateral cell groups of the S2-S4 spinal segments, releasing acetylcholine that acts on M3 muscarinic receptors on the detrusor muscle, causing its contraction and facilitating urine expulsion. At the same time, projections from the PMC to the GABAergic and glycineergic interneurons in the spinal cord lead to inhibition of the Onuf nucleus in the anterior horn cells of the S2-S4 segments, causing relaxation of the striated external urethral sphincter. This ultimately allows voiding to occur (Fig. 2B) (*Arya & Weissbart, 2017*; *Fowler, Griffiths & De Groat, 2008*; *Karnup, 2021*; *Kim et al., 2020*; *Yoshimura & De Groat, 1997*).

# RESEARCH METHODS FOR LUT NEURAL REGULATION MECHANISMS

## Electrophysiological recording

Neural control of behavior is achieved through computation, in which neurons integrate electrical signals and generate electrical impulses as output. Electrophysiology allows for the recording of crucial signals that control the activity and function of neural systems, tissues, and organs, making it highly suitable for capturing the natural language of the brain. Electrophysiological techniques combine high spatiotemporal resolution with ease of application, making them particularly attractive tools for studying awake, behaving subjects. By analyzing the electrical activity of neurons through electrophysiological techniques, the mechanisms of neural regulation can be revealed. Furthermore, electrophysiological signals can also be used as command signals in human-machine interfaces, holding promise for significant contributions in rehabilitation and cognitive enhancement (*Chorev et al., 2009*; *Zhu et al., 2021*).

Electrophysiological recording techniques are based on the electrical activity of neurons and primarily include intracellular recording and extracellular recording methods. Intracellular recording involves inserting a microelectrode into the cell membrane of individual neurons to record their membrane potential changes. This technique provides high-resolution electrical signal recordings that can capture the details of small intracellular voltage changes. Consequently, it has been widely used in behavioral experiments with awake animals, allowing the monitoring of intrinsic characteristics of neurons during different global states and learning processes. Such data contribute to linking the knowledge of cellular discharge patterns in behaving animals with cellular mechanisms typically studied in *in vitro* preparations. However, the success rate of intracellular recording in awake animals is lower and is limited to a few cells. Additionally, due to the potential impact of electrode insertion on cell stability, intracellular recordings are usually performed over a short time period (*Chorev et al., 2009*; *Lee & Brecht, 2018*; *Long & Lee, 2012*; *Wickenden, 2014*). In contrast, extracellular recording is a non-invasive technique that involves placing electrodes in the surrounding tissue or fluid medium of neurons to record the electrical signals generated by neuronal firing. This method avoids direct trauma to cells. Moreover, the electrodes can receive signals from multiple surrounding cells, providing observations of cellular population activity. There are various variants of extracellular recording techniques, which can be classified into single-point electrodes and multi-site electrodes. The advantage of multi-site electrodes lies in their ability to reliably separate signals between multiple recording points through triangulation, thereby achieving the separation of multi-unit signals. Extracellular recording has already taken a dominant position in the field of systems neuroscience and will continue to play a crucial role (*Chorev et al., 2009*). However, it should be noted that due to the distance between the electrode and the cells, extracellular recording cannot capture the details of small intracellular voltage changes. Furthermore, the received signals may originate from the activity of multiple surrounding cells, potentially biased towards more active and larger neurons, leading to

significant sampling biases (*Chorev et al., 2009*; *Montagni et al., 2019*; *Sun & Zhao, 2021*; *Wickenden, 2014*).

Electrodes, as the key components for collecting electrophysiological signals, directly influence the quality and reliability of the signals. Depending on the application scenario, electrodes are generally classified into invasive and non-invasive types. Invasive electrodes are inserted into the biological tissue to record internal electrical activities, enabling precise signal acquisition with high spatial and temporal resolution. Common types include non-penetrating thin-film microelectrode arrays wrapped around internal organ surfaces and penetrating electrodes that traverse three-dimensional tissues. Non-invasive skin electrodes allow convenient signal collection from the skin without causing harm to the human body (*Zhu et al., 2021*). They come in various forms, such as surface patch electrodes and tattoo-like electrodes (*Lopes et al., 2019*; *Wang et al., 2020*). Since electrodes used for measuring electrophysiological signals must directly contact the body, both invasive and non-invasive types should possess good biocompatibility, minimizing skin irritation and allergic reactions in external usage and reducing foreign body response in internal usage (*Kim et al., 2018*). Although significant progress has been made in recent years with the development and application of flexible organic electronic materials, biomaterials, and nanomaterials for electrophysiological signal acquisition, there are still several unresolved challenges that warrant further attention (*Schendel, Eliceiri & Williams, 2014*). For instance, the durability of flexible electrodes requires further evaluation, and there is a need to improve the signal-to-noise ratio of recorded electrophysiological signals (*Zhu et al., 2021*). Additionally, the *in vivo* delivery of nanomaterials may present a larger design challenge, as the surface chemical properties, size, shape, and other factors of nanoscale particles can influence their ability to target specific regions of interest and their residence time at neural interfaces (*Chen, Canales & Anikeeva, 2017*).

Electrophysiological techniques are capable of accurately recording the temporal and spatial characteristics of neuronal electrical activity, enabling real-time monitoring of neuronal activity and capturing dynamic changes in interactions and network dynamics among neurons. Thus, electrophysiological measurements have remained the most valuable approach in studying sensory, motor, and cognitive phenomena in the human nervous system. They serve as a valuable tool for characterizing human health and find extensive applications in fields such as medicine, engineering, and pharmaceuticals. Research on electrophysiological signals primarily focuses on the precise acquisition, real-time monitoring, and interpretation of these signals. Long-term continuous monitoring of human biological signals, such as electrocardiography (ECG), electromyography (EMG), electroencephalography(EEG), and electrooculography (EOG), enables early medical diagnosis and rehabilitation of diseases. Through electrophysiological recordings, we can investigate abnormal patterns of neuronal activity, understand neural circuit abnormalities associated with diseases, and provide guidance for disease diagnosis and treatment (*Kamarajan & Porjesz, 2015*; *Lee, Park & Yoo, 2022*). Currently, this technique has been widely applied in the study of LUT neurons to explore related issues such as LUT dysfunction (*Karnup & De Groat, 2020*; *Kawatani et al., 2021*; *Ma et al., 2022*; *Verstegen et al., 2019*). Researchers can utilize electrophysiological recording techniques to capture

the electrical signals of LUT tissues, such as the bladder and urethra. These signals can reveal the discharge patterns, activity frequencies, and interactions among neurons. Furthermore, electrophysiological recording can be combined with other techniques, such as optogenetics or pharmacological experiments, to further investigate the molecular and cellular mechanisms of LUT neural regulation. By integrating these techniques, we can gain a deeper understanding of the complexity of LUT neural regulation and provide new insights and strategies for the treatment and management of related diseases. A review by *Curt & Dietz (1999)* also confirms the reliability of combining clinical and electrophysiological recording in predicting the degree of upper and lower limb functional recovery in patients with acute spinal cord injury. *Gross et al. (2016)* have also demonstrated the potential of transcutaneous electrical nerve stimulation (TENS) for treating neurogenic lower urinary tract dysfunction (NLUTD) through a systematic review. By combining appropriate examination techniques, we can predict the recovery of hand function, walking ability, bladder function, and autonomic nerve function. Additionally, the application of clinical and electrophysiological recording can assist in assessing the spontaneous recovery of the spinal cord and evaluating new treatment methods (*Curt & Dietz, 1999*).

In conclusion, electrophysiological recording has significant value in the study of LUT neural regulation. It can provide quantitative and qualitative information about neuronal electrical activity, helping us gain a deeper understanding of the function and disease mechanisms of the LUT.

## Optogenetics

Optogenetics is an emerging technology that integrates methods from the fields of genetics and optics, enabling optical control of specific populations of neurons with high temporal and spatial resolution. By leveraging the properties of light, this technique alters cellular behavior without affecting other neurons in the brain. Compared to conventional electrophysiological techniques and pharmacological methods, optogenetics offers greater selectivity and specificity (*Yao, Hou & Yin, 2012*). Due to its ease of use and high spatiotemporal precision, optogenetics has found widespread applications in various domains of neuroscientific research. For instance, it has been employed in investigations of somatosensory circuits, visual circuits, auditory circuits, olfactory circuits, as well as studies on neurological disorders. Optogenetics serves as a powerful tool for exploring the mechanisms and functions of these neural circuits and provides valuable insights into the pathophysiological processes underlying certain neurological diseases (*Neghab et al., 2022*).

Optogenetics has profoundly transformed neuroscience and increasingly provides opportunities to understand the role of specific neural circuits in physiological behavior. By using appropriate optogenetic proteins and manipulating the activity of specific brain region neurons associated with neurological disorders, the pathology of the nervous system diseases can be identified, and new therapeutic approaches can be developed. Most neuroscience studies utilize light-sensitive proteins called opsins. These photosensitive actuators are transmembrane domain proteins that undergo conformational changes when exposed to visible light within a specific wavelength range (390–700 nm), resulting

in ion currents on the cell membrane, causing depolarization or hyperpolarization of the cells (*Neghab et al., 2022*). In optogenetics, there are many different types of opsins available, with the most widely used being the channelrhodopsin-2 (ChR2), which has been mammalian codon optimized, and the third-generation enhanced halorhodopsin (eNpHR3.0), which has been optimized for mammalian cell membrane trafficking (*Aston-Jones & Deisseroth, 2013*). When expressed in neurons, ChR2 produces a blue light-sensitive cation channel, which opens upon blue light activation, allowing Na+ ions to enter the neuron and causing excitatory depolarization. In contrast, NpHR is a Cl- ion pump protein that binds to the neuronal cell membrane and is activated by yellow light. Upon activation, the pump moves Cl- ions into the neuron, hyperpolarizing the cell membrane and inhibiting neural activity (*Montagni et al., 2019*; *Neghab et al., 2022*).

To achieve selective expression of opsin genes in specific or chosen types of neurons in the brain, one of the most effective methods is to utilize viruses as gene delivery vectors. Viruses deliver the genetic material encoding opsins into cells such as neurons through a mild transformation process to achieve *in vivo* expression (*Gerits & Vanduffel, 2013*). Currently, three commonly used viral delivery methods include lentivirus, adeno-associated virus (AAV), and Cre-dependent AAV expression systems. Lentivirus is a widely used delivery tool that can transfer genetic material into specific or chosen types of neurons in the brain and allows for selective high-level expression of opsin genes. The optimal gene expression time for lentivirus is approximately two weeks after injection and can last for several years. The other two commonly used methods are delivering genetic material to specific types of neurons through AAV and Cre-dependent AAV expression systems, with the optimal gene expression time being three weeks after injection. However, a general limitation of viral vectors is their limited packaging capacity, restricting the use of relatively small, specific, and efficient genes for delivery (*Altahini, Arnoux & Stroh, 2024*; *Neghab et al., 2022*). Therefore, when selecting appropriate opsins and delivery methods, factors such as target cell type, gene size, and expression requirements need to be considered comprehensively.

Optogenetics was initially used solely for controlling neural circuits in the brain. However, it has been employed to regulate the activity of the spinal cord and peripheral nervous system. In particular, the application of optogenetics in bladder storage and micturition regulation has garnered significant attention in recent years (*Mondello et al., 2023*; *Park et al., 2017*; *Zhou & Liao, 2021*). *Kelley et al. (2016)* were the first to demonstrate the use of optogenetics to modulate LUT sensation. Their findings indicated the feasibility of optogenetic modulation of peripheral nerves through stimulation of the sciatic nerve. This work represents an initial exploration of optogenetics in LUT applications and underscores its utility in research on neural control of the LUT (*Kelley et al., 2016*). Optogenetics, when applied to the study of urinary-related neural mechanisms, allows for the selective activation or inhibition of specific types of neurons by choosing appropriate optogenetic proteins and stimulation parameters. This enables the investigation of their functions and interactions in urinary control. For instance, *Yao et al. (2018)* demonstrated through the use of optogenetics and other techniques that a small population of layer 5 (L5) neurons in the primary motor cortex (M1) plays a crucial role in descending control of micturition

by projecting to the pontine micturition center (PMC). This discovery provides new insights into the cortical mechanisms of urinary control (*Yao et al., 2018*). Furthermore, optogenetics can be widely applied in the study of neural mechanisms underlying bladder pain and pain transmission, providing a crucial foundation for the development of new treatment methods and interventions. Research by *Park et al. (2017)* confirmed that optogenetic modulation of smooth muscles is an effective means of actively controlling bladder contractions with spatial and temporal precision. This modulation approach holds promise for improving bladder control in the treatment of LUT dysfunction while avoiding the side effects of conventional clinical therapies. Studies by Sarah E. Mondello and colleagues also demonstrated that neuron-specific optogenetic spinal cord stimulation significantly promotes the recovery of skilled forelimb extension. Additionally, the labeling of GAP-43 and laminin significantly increased in the optochemical stimulation group, indicating the promotion of axonal growth and angiogenesis. These findings suggest that optogenetic stimulation is a potent neuromodulatory tool that may play a role in future therapies and research aiming to investigate the role of specific cell types, pathways, and neuron populations in the recovery after spinal cord injury (*Mondello et al., 2023*).

In conclusion, optogenetics provides a powerful tool for studying the neural mechanisms of the LUT and serves as a crucial foundation for the development of new treatment methods and interventions.

## METHODS FOR LOWER URINARY TRACT FUNCTION EVALUATION

By observing and conducting experimental studies on the urination behavior of rodents, we can greatly enhance our understanding of LUT function and gain deeper insights into the complex social, environmental, and internal stimulus factors that influence urination in health and disease models (*Wang et al., 2019*). Currently, non-invasive examinations and more invasive urodynamic assessments are commonly employed to evaluate the LUT function in animals, particularly rodents. Non-invasive examinations provide preliminary insights into LUT function. On the other hand, urodynamic assessments represent the sole method for objectively assessing LUT function, enabling differentiation of various pathologies and investigation of potential neuronal functional impairments (*Sartori, Kessler & Schwab, 2021*). A comprehensive understanding of these methodologies facilitates researchers in better observing and documenting the characteristics of animal urination behavior, thereby enhancing the comprehension of LUT function and neural regulatory mechanisms.

### Noninvasive assays

Non-invasive examination methods primarily utilize relatively simple tests to assess the urination frequency and volume in animals, and in certain cases, the flow rate can also be measured. Overall, non-invasive examination methods provide preliminary information for evaluating the LUT function in animals, offering the advantage of simplicity in operation (*Sartori, Kessler & Schwab, 2021*).
## Principles and applications of void-spot assay

The void spot assay (VSA), also known as voiding spot on paper (VSOP) assay, is a non-invasive testing method. It involves placing mice in shells lined with absorbent filter paper and allowing them to freely move within a designated time frame. During the testing process, urine spots on the filter paper are imaged and analyzed quantitatively using automatic fluorescence emitted by the urine under ultraviolet light. Additionally, urine spots can be visualized using ninhydrin spray and imaged and quantified under bright field illumination (Fig. 3A). VSA provides cumulative information about mouse urination behavior over time, similar to the information provided by urination diaries in human medicine. Mouse urination is a spontaneous behavior influenced by bladder fullness and social behavior (*Chen et al., 2017*; *Hill et al., 2018*). Currently, VSA methods are widely used to evaluate and quantify urination function in rodents. For example, *Keller et al. (2018)* utilized VSA to quantitatively analyze mouse urination behavior in studies on the autonomous urination neural circuit. *Ruetten et al. (2021)* demonstrated that VSA can differentiate urination behavior between different experimental groups, making it applicable to defining urination patterns in male mice with diabetes insipidus. This study also provided the first evidence that VSA can distinguish different types of functional impairments.

The greatest advantage of VSA is its non-invasiveness, allowing animals to remain intact for other purposes. Furthermore, VSA is cost-effective, easy to implement and execute, and can be repeatedly used to track changes in urination function over time. However, this method also has limitations. Firstly, without a synchronized video recording system, overlapping urination events may lead to biased analysis results. Secondly, the urination patterns of mice during VSA may be influenced by various factors, including cage type, cage floor, water availability, water bottle placement, housing conditions (individual or group), and different handlers. Additionally, VSA experiments require manual input and offline processing, resulting in a lack of real-time analysis results (*Chen et al., 2017*; *Sartori, Kessler & Schwab, 2021*). To address the current issues associated with the VSA method, several studies have proposed improvements based on VSA. For instance, Marianela G. Dalghi and colleagues developed a method called real-time VSA (RT-VSA) video monitoring to overcome the limitations of temporal resolution and quantification of overlapping urine spots in VSA as an endpoint analysis. This cost-effective approach provides temporal, spatial, and volumetric information about urination events during light and dark phases of the day, along with details related to mouse behavior (*Dalghi et al., 2023*). Furthermore, *Luo et al. (2023)* investigated the sensitivity of VSA results to housing conditions and procedural parameters, aiming to evaluate the comparability of VSA results across different laboratories by minimizing these variables. They found that the analysis tools between Fiji and MATLAB demonstrated good consistency in quantifying VSA parameters, particularly the primary void spot (PVS) parameter. This study provided insights into the importance of minimizing variables to achieve comparability of VSA data across laboratories, emphasizing the significance of standardized transportation, acclimation, and time of day for generating consistent VSA results. These novel improvements and insights

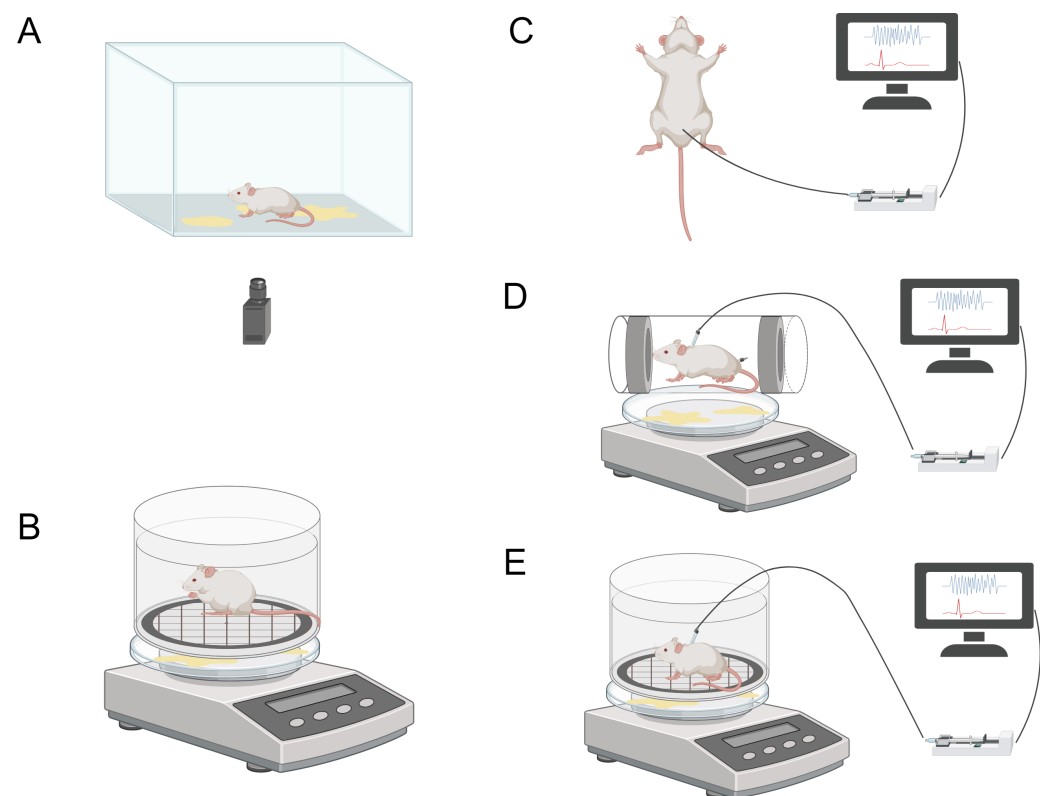

**Figure 3** **Schematic diagram illustrating various methods for the assessment of lower urinary tract function.** The image components are sourced from https://www.biorender.com/features. (A) Void-spot assay. (B) Metabolic cage. (C) Urodynamics in anesthetized animals. (D) Urodynamics in restrained animals. (E) Urodynamics in freely moving animals.

may contribute to enhancing the quality and analysis efficiency of VSA data, ultimately leading to better scientific conclusions (*Luo et al., 2023*).

## Principles and applications of metabolic cages

There are various commercially available metabolic cages on the market for collecting urine from experimental animals. Typical metabolic cages are constructed from transparent and bite-resistant polycarbonate material, with a removable ventilated lid for accommodating the animals. Additionally, there is a feeding chamber outside the metabolic cage, containing a sliding drawer for convenient addition of feed, liquid, or powder without disturbing the animals. This setup prevents feed contamination of the urine. Furthermore, the external space of the metabolic cage is designed to be small, discouraging nesting or sleeping behaviors of the animals. The design of the water bottle holder and overflow collection tube aims to prevent water from entering the cage and contaminating the urine (Fig. 3B). The overflow collection tube is calibrated to allow researchers to accurately measure the actual water consumption of the animals. By using a digital scale beneath the grid to collect urine, it is possible to determine urine volume, voiding time, flow rate, and voiding frequency (*Kurien, Everds & Scofield, 2004*; *Sidler et al., 2018*).

The advantages of using metabolic cages for urine assessment include the ability to conduct 24-hour measurements, thereby recording the animals' voiding patterns during both active and resting phases. However, the metabolic cage method has certain limitations. Firstly, it is difficult to accurately diagnose potential pathological and physiological conditions. Additionally, for very small urine samples, there may be adsorption onto the grid, limiting the precision of measurements. Moreover, keeping the animals in a new and empty environment for extended periods may induce stress, which could potentially affect their voiding behavior (*Kurien, Everds & Scofield, 2004*; *Sartori, Kessler & Schwab, 2021*). Furthermore, existing research has demonstrated measurable effects of spontaneous and induced behaviors in laboratory rats due to the metabolic cages. These behavioral changes may lead to negative emotional states in these animals, resulting in adverse consequences. Therefore, further research is needed to quantify the presence and extent of these impacts on the health of experimental animals (*Whittaker, Lymn & Howarth, 2016*).

### Invasive assays

Urodynamics is primarily a dynamic study of the transport, storage, and voiding of urine within the urinary tract. It involves multiple interacting tests, including uroflowmetry, electromyography (EMG), cystometry, pressure-flow studies (PFS), leak point pressure measurement, post-void residual (PVR) measurement, and urethral pressure profilometry. The evaluation can encompass all urodynamic components or be limited to specific elements to address specific questions. Urodynamic examination provides a comprehensive assessment of LUT system function and can differentiate different pathological changes and investigate potential neuronal functional disorders. Currently, invasive urodynamic measurements in rodent animals primarily consist of the following three types (*Brown, Krlin & Winters, 2013*; *Rutman & Blaivas, 2007*).

### Intravesical pressure measurement

The bladder pressure measurement method is a technique for measuring the pressure inside the bladder. It involves inserting a catheter into the bladder, filling it with a physiological solution, and connecting it to a pressure sensor to measure the pressure changes within the bladder. The pressure sensor converts the bladder pressure into an electrical signal, which can be recorded and analyzed to provide quantitative data about bladder pressure, evaluating bladder contractility, compliance, and voiding function.

During bladder pressure measurement, conscious animals can be chosen for experimentation and placed in a metabolic cage (Fig. 3E). Another option is to place awake and habituated animals in a restrainer and induce voiding by filling the bladder through a catheter within 1–3 hours (Fig. 3D). Restraining the animals can reduce motion artifacts, but it is crucial to acclimate and habituate the animals to the restraint to minimize stress-related changes in voiding behavior (*Schneider et al., 2015*). Urine can be collected on filter paper and weighed using a scale connected to the sensor. The bladder catheter is connected to the pressure sensor and infusion pump through a T-shaped connector or three-way stopcock. Two sensors are connected to a sensor amplifier, which is then connected to a data acquisition system to track and record bladder pressure and urine volume during the experiment, using specific software for storage and further analysis.

Although human urodynamics studies are typically performed in a conscious state without anesthesia to mimic daily life conditions as closely as possible, in animal research, it is more common to conduct urodynamic investigations under anesthesia to minimize motion artifacts (Fig. 3C) (*Brown, Krlin & Winters, 2013*). For bladder pressure measurement in anesthetized animals, the same system can be used, but the animals need to be placed in a supine position, preferably on a temperature-controlled animal blanket, with urine collected in a urine collection cup. However, it is currently uncertain whether the supine position affects bladder measurement parameters. Furthermore, studies have shown that while aminocaproic acid seems to be the preferred anesthetic for maintaining voiding reflexes, it severely impairs bladder function, leading to significant differences in urodynamic results compared to the awake state. Therefore, careful selection of appropriate anesthetics is necessary in animal urodynamic studies to ensure the accuracy and reliability of measurement results (*Andersson, Soler & Fuellhase, 2011*; *Sartori, Kessler & Schwab, 2021*).

## Leak point pressure measurement

Leak point pressure measurement (LPP) is also a method for evaluating bladder function, which indirectly reflects changes in bladder pressure by measuring the pressure of urine leakage from the urethra. This measurement method is commonly used in the study of urinary control and voiding behavior in animals. Leak point pressure measurement can be divided into two types: bladder leak point pressure (BLPP) and abdominal leak point pressure (ALPP). In comparison to the direct measurement of bladder pressure in intravesical pressure measurement, the main difference between leak point pressure measurement and intravesical pressure measurement lies in the location of catheter insertion and the different measurement targets. Leak point pressure measurement is an indirect method of measuring bladder pressure, where BLPP is measured by inserting a catheter into the urethra of mice or animals to measure the pressure before urine leakage, while ALPP is measured by inserting a catheter into the abdomen of mice or animals to measure the pressure of the abdominal muscles, thereby indirectly reflecting changes in bladder pressure. However, this indirect measurement method may also lead to inaccuracies or errors in the data (*Kelly & Krane, 2000*).

## External urethral sphincter electromyography

Electromyography (EMG) in urodynamic studies is used to assess the coordination or lack of coordination between the external sphincter and detrusor muscle. By recording and analyzing the electromyographic activity of the sphincter muscle, EMG provides information about muscle contraction and relaxation, enabling the evaluation of sphincter function and activity patterns. During EMG measurements, electrodes are placed near the external sphincter of an anesthetized or awake animal, ensuring close contact with the muscle to accurately record muscle electrical activity. Subsequently, the electrodes are connected to an electromyograph or a bioelectric amplifier to record the electromyographic signals of the sphincter muscle, with the duration of signal recording adjusted as needed

(*Kelly & Krane, 2000*). The recorded electromyographic signals can be subjected to time-domain analysis and frequency-domain analysis to calculate the amplitude, frequency, and temporal characteristics of muscle activity (*Alcan & Zinnuroglu, 2023*).

Functional assessment of the EUS in mouse urodynamic studies presents a challenge. Due to its small size, there are technical difficulties in examining the EUS. Therefore, compared to studies in rats, a comprehensive characterization of bladder and urethral coordination activity in normal and pathological conditions in mice has not been achieved. There have been limited reports on EUS-EMG measurements in awake mice. Some reports indicate the presence of low-amplitude sustained EUS-EMG activity during the inter-micturition interval. However, these reports primarily focus on EUS-EMG activity during the voiding phase, and the specific details of the voiding phase remain incompletely described (*Ito et al., 2017*).

In summary, although urodynamic studies provide a comprehensive and objective assessment of LUT function, they are more expensive and time-consuming compared to non-invasive detection methods. Additionally, invasive urodynamics involve catheterization and bladder filling, which may influence voiding patterns (*Sartori, Kessler & Schwab, 2021*). To overcome the limitations of invasive experiments, efforts are being made to develop methods for assessing lower urinary tract voiding function in awake, freely moving animals. For example, *Verstegen et al. (2020)* developed a novel non-invasive detection method called the Micturition Video Thermography (MVT), which uses a downward-facing thermal camera above mice on filter paper flooring. MVT is a reliable and non-invasive method for measuring voiding time, voided volume, and voiding location, and unlike bladder imaging preparations, MVT does not require surgical catheterization. Additionally, *Mickle et al. (2019)* developed an optogenetic peripheral nerve modulation wireless closed-loop system that utilizes a soft strain gauge to obtain real-time bladder functional information in a rat model, with data algorithms capable of identifying pathological behavior.

## APPLICATION AND PROSPECTS OF ARTIFICIAL INTELLIGENCE IN LUT NEUROMODULATION RESEARCH

### Machine learning and deep learning

Machine learning (ML) is a type of artificial intelligence (AI) technology that has rapidly developed since the 1990s and has become one of the most successful areas of AI. It is widely applied in fields such as image and speech recognition, traffic alert systems, autonomous vehicles, and medical diagnostics (*Mostafa, Zisis & Moustafa, 2022*). ML trains computers to make predictions based on available datasets and algorithms by learning from past experiences and expanding upon them. It empowers computer systems with the ability to learn and improve themselves, reducing the need for human intervention by analyzing data and discovering trends. It primarily utilizes statistical and optimization methods to construct models and find patterns and regularities in the data. Machine learning encompasses various algorithms including decision trees, support vector machines (SVM), random forests, and naive Bayes, among others. These algorithms achieve tasks such as

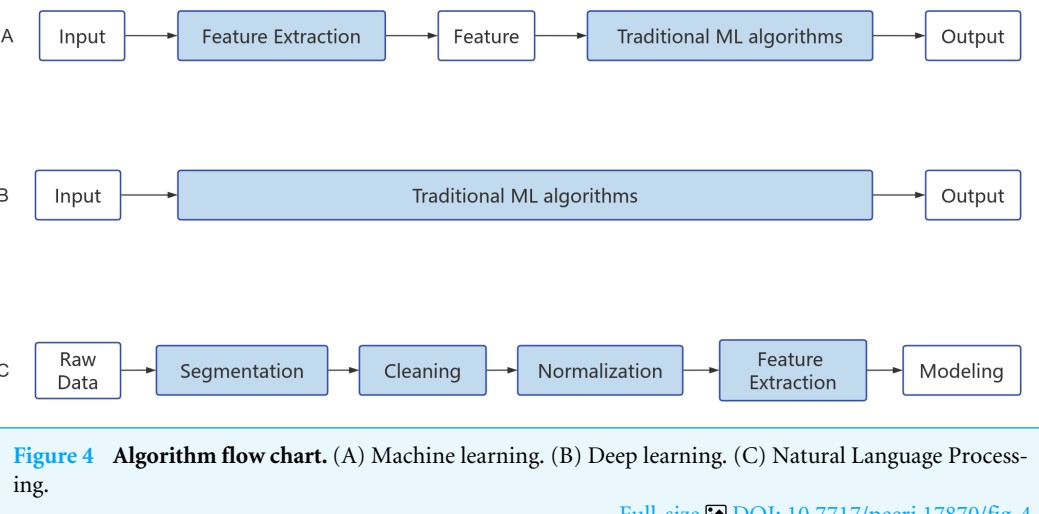

**Figure 4  Algorithm flow chart.** (A) Machine learning. (B) Deep learning. (C) Natural Language Processing.

data classification, regression, and clustering through feature extraction and model training (*Mahadevkar et al., 2022*).

With the rapid development of technologies such as cloud computing, big data, and the Internet of Things, the emergence of various applications in cyberspace has led to an explosive growth in data volume. Big data holds rich value and enormous potential, bringing about transformation and development in human society, but it also presents the challenge of information overload. The efficient extraction of valuable information from complex and abundant data has become a critical challenge (*Mu & Zeng, 2019*). Deep learning (DL), a subfield of machine learning, is a powerful method for analyzing big data. It simulates the structure and functionality of the human brain's neural network and uses multi-layered neural networks for learning and decision-making. DL can automatically learn features from heterogeneous data sources, mapping different types of data to a unified latent space to obtain a unified data representation (*Talaei Khoei, Ould Slimane & Kaabouch, 2023*). The main difference between deep learning and traditional machine learning algorithms is that traditional machine learning typically relies on manually selecting and extracting features to represent data, while DL automatically learns feature representations through stacked neural networks. The workflow of traditional ML and DL algorithms is illustrated in the figure below (Figs. 4A–4B).

ML and DL, as core technologies of artificial intelligence, play an important role in various applications such as detection, classification, and segmentation in the field of healthcare. Current research has demonstrated that ML and DL also have significant application value in the study and diagnosis of LUT neuromodulation (*Pang, Li & Zhao, 2023*; *Velliangiri et al., 2019*). By learning from and pattern recognition of a large amount of data, these techniques can help us gain a deeper understanding of the complexity of LUT neuromodulation and provide predictive capabilities for LUT dysfunction. Firstly, ML and DL can be applied to the observation and experimental research of urination behavior in rodents. Behavioral recognition in mice is widely applied in research fields such as biology, neuroscience, and pharmacology, providing a basis for assessing the psychological

and physiological states of mice. Traditional behavioral analysis methods mostly rely on manual observation and sensors, which are time-consuming, labor-intensive, and prone to biases. Moreover, the influences caused by sensors and manual observation, such as sound, light, electricity, and odor, can interfere with the natural behavior of experimental animals, leading to biased experimental data. To address this issue, *Liu, Zhu & Rao (2022)* proposed a symmetric algorithm based on temporal and spatial information capture of behavioral changes. This method demonstrates high accuracy in key point detection and behavioral recognition, providing a useful tool for mouse movement behavior recognition. *De Bruyn et al. (2022)* also developed a ML approach to address the inability of videocystometry to be implemented in awake and freely moving animals. By training a neural network to automatically detect a large number of delayed perspective images of the mouse bladder, they accurately determined relevant parameters such as bladder volume and flow rate. This study successfully identified the profound impact of urethane (an injection anesthetic standardly used in preclinical bladder research) on LUT function by combining videocystometry and ML. Additionally, the combination of perspective imaging and machine learning-based image analysis revealed the significant influence of micturition on bladder functional capacity (*De Bruyn et al., 2022*). Furthermore, ML and DL can also be applied to vast physiological data generated in experimental research, including urine composition and urodynamic parameters. For instance, addressing the issues of time-consuming manual analysis and potential accuracy concerns, *Liang et al. (2018)* proposed applying convolutional neural networks to urinary sediment examination (USE). By automating the auxiliary examination process, they effectively overcame the current reliance on manual microscopy examination, significantly improving diagnostic efficiency.

Secondly, ML and DL can be used for the observation of patients' urinary behavior in a clinical setting. *Na & Kim (2021)* developed a support vector machine-based algorithm for identifying female voiding intervals, which can recognize the timing and intervals of voiding based on specific postures and posture changes of patients. This voiding behavior recognition technology exhibits high accuracy and can be applied in clinical settings to characterize the voiding patterns of female patients. Recent reports on ML algorithms have also indicated their accuracy in identifying overactive detrusor based on tracking, insufficient detrusor activity, and other urodynamic phenomena. *Knorr & Werneburg (2024)* also affirmed in a review that ML may soon realize its potential by integrating clinical factors, diagnostic data (including urodynamic data), and patient preferences to optimize diagnosis and tailor clinical management for specific patients. Lastly, ML and DL can be used to detect and intervene in LUT dysfunction. By training and pattern recognition of a large amount of urinary tract neuromodulation data, these techniques can learn the features and patterns of normal urinary tract function and detect abnormal conditions. This provides new means for the early detection of urinary tract dysfunction, contributing to the improvement of diagnostic capabilities and treatment effectiveness in urinary tract diseases. For example, researchers have utilized data algorithms to identify pathological behaviors and normalize bladder function through automatic closed-loop optogenetic neural modulation of bladder sensory input (*Mickle et al., 2019*).
## Natural language processing

Natural language processing (NLP) is an important branch of research and application in the field of artificial intelligence. Its goal is to enable computers to understand, analyze, and generate human language. NLP integrates knowledge from various disciplines such as computer science, artificial intelligence, and linguistics, with the aim of building intelligent systems that can process and comprehend natural language. In recent years, NLP has experienced rapid growth due to the increasing availability of textual data and the demand for more complex and human-like communication between computers and humans. The importance of NLP lies in its ability to transform the way humans and computers interact, making their communication more intuitive and human-friendly. This has wide-ranging practical applications in areas such as information retrieval, machine translation, and question answering (*Cambria & White, 2014*; *Patwardhan, Marrone & Sansone, 2023*).

NLP techniques primarily consist of rule-based methods and ML methods. Rule-based methods involve the use of predefined rules, such as regular expressions, word matching, and integrated annotations, to select the text to be retrieved or synonyms. On the other hand, ML methods typically require a large amount of text and datasets for training, validation, and testing. When using ML methods, it is necessary to preprocess the text for NLP processing and select suitable algorithmic models. Common models include classical ML methods like random forests, models utilizing vectorized statistical methods, as well as DL models such as convolutional neural networks (CNN) for text classification, recurrent neural networks (RNN) for sequence labeling, and Attention mechanisms for machine translation (Fig. 4C). NLP has been widely applied in various clinical environments, with extensive research conducted particularly in the fields of mental health, breast cancer, and pneumonia (*Pons et al., 2016*; *Yang et al., 2022*). For example, *Yang et al. (2022)* developed a large-scale clinical language model called GatorTron, trained on a corpus of over 90 billion words, including more than 82 billion words of unstructured clinical text. Expanding the parameters of the clinical language model from 110 million to 8.9 billion, the GatorTron model demonstrated improvements in five clinical NLP tasks and can be applied to medical artificial intelligence systems to enhance healthcare services. These studies have also demonstrated how NLP techniques aid in the extraction, organization, and analysis of medical information from extensive literature and clinical records (*Yang et al., 2022*). Furthermore, they highlight how NLP can provide new methods for studying the LUT neural regulatory mechanisms. Through text mining and knowledge graph construction, the discovery and accumulation of knowledge regarding LUT neural regulation-related research can be accelerated.

## Data mining and pattern recognition

Data mining is the process of cleaning, processing, analyzing, and extracting useful information and patterns from large-scale datasets. It utilizes techniques such as statistics, machine learning, and pattern recognition to reveal hidden knowledge and information by analyzing the features, associations, and trends within the data (*Gul, Bano & Shah, 2021*). On the other hand, pattern recognition involves processing data through classification, clustering, identification, and prediction to discover and identify patterns, regularities,
and structures within the data. The main components of a pattern recognition system include data collection and acquisition, feature extraction and representation, similarity detection and pattern classifier design, and performance evaluation (*Jain, Duin & Mao, 2000*; *Rosenfeld & Wechsler, 2000*).

Data mining and pattern recognition have enormous potential for development in the research of LUT neuromodulation. By utilizing these techniques, it is possible to analyze patients' clinical data, physiological signals, and imaging data to assist in identifying and classifying different diseases. Additionally, it helps to recognize individual differences and treatment responses among patients, enabling personalized neuromodulation therapy for the LUT. Previous studies have combined process mining with interactive pattern recognition (IPR) methods to support healthcare professionals in deploying personalized care plans for cardiovascular diseases (*Fernandez-Llatas et al., 2016*). Furthermore, through the application of data mining and pattern recognition to analyze data obtained from studies on the neuroregulatory mechanisms of the LUT, it is possible to delve into the regulatory role of brain regions in the urination mechanism. Such analysis can reveal the associations between different brain regions and LUT functions, identify neural patterns and signal characteristics related to urine control, thus enhancing our understanding of neuroregulatory mechanisms. In-depth research on these patterns and mechanisms can provide new perspectives and potential therapeutic strategies for the treatment of urinary control disorders and related diseases. With further advancements in data collection and storage technologies, it will be possible to obtain more comprehensive data related to LUT neuromodulation, providing a richer source of information for data mining and pattern recognition. Additionally, by integrating emerging technologies such as DL and neural networks, it is possible to construct more complex and accurate models, further enhancing the diagnosis and treatment of neuroregulatory diseases of the LUT (*Djellali & Adda, 2019*). Furthermore, further research is needed to address issues such as data privacy protection, algorithm interpretability, and validation in clinical practice to ensure the reliability and applicability of data mining and pattern recognition in LUT neuromodulation research.

## Intelligent assisted diagnosis and treatment system

The Intelligent Assisted Diagnosis and Treatment System is a system that employs artificial intelligence and related technologies to aid physicians in disease diagnosis and treatment decision-making. This system has the capability to analyze and interpret vast amounts of medical data, providing accurate diagnostic recommendations and personalized treatment plans, thereby enhancing medical efficiency and accuracy (*Manickam et al., 2022*; *Mann, 2023*). Presently, research on intelligent assisted diagnosis systems has encompassed various aspects. For instance, *Zhang (2015)* conducted a study on intelligent knowledge-based assistance in internal medicine diagnosis, building upon previous research. They utilized knowledge bases and a mathematical model called analytic hierarchy process to assist in patient treatment. The hierarchical model can summarize disease characteristics and quantify diagnostic criteria. By combining physician judgment on the weights of various factors in the criteria layer and comparing databases, the weighted value for a potential disease can be determined. Physicians can leverage this system to rapidly complete

diagnoses, utilizing the quantitative values of the mathematical model to facilitate the disease matching process and provide assistance to physicians (*Zhang, 2015*). Furthermore, a review by *Li et al. (2020)* highlighted the tremendous prospects of AI-assisted diagnosis systems in the field of pediatrics.

In the realm of LUT neurophysiology research, the Intelligent Assisted Diagnosis and Treatment System may also play a crucial role. Firstly, it can aid physicians in more accurately diagnosing diseases related to LUT neurophysiology, such as bladder dysfunction and urinary incontinence. The system can analyze and reason patient's clinical data, imaging materials, and laboratory test results, in conjunction with extensive medical knowledge bases, to provide comprehensive disease assessment and diagnostic recommendations. Moreover, the Intelligent Assisted Diagnosis and Treatment System can establish personalized treatment models and prediction models, offering guidance and predictions for LUT neurophysiology research. By utilizing machine learning and data analysis techniques based on patient characteristics and medical data, the system can construct prediction models to forecast disease progression trends and treatment outcomes, thereby providing decision support and optimizing treatment strategies for researchers.

## CONCLUSIONS

Currently, significant progress has been made in studying the neural regulatory mechanisms of the LUT using techniques such as electrophysiological recording and optogenetics, revealing some of the interactions and regulatory mechanisms among relevant neurons. However, the LUT involves various types of neurons and neurotransmitters, and understanding the interactions and regulatory networks among neurons requires extensive data analysis and integration, necessitating the support of emerging technologies. With the development of emerging technologies like artificial intelligence, we also see vast prospects for studying the neural regulatory mechanisms of the LUT. Artificial intelligence is expected to play a role in the treatment of LUT dysfunction by providing more precise and effective treatment plans through personalized predictive models and intelligent assistance systems.

In summary, although there are challenges and difficulties in studying the neural regulatory mechanisms of the LUT, emerging technologies offer new opportunities. Through the integration of various technical approaches, we have the potential to gain a deeper understanding of the mechanisms underlying LUT neural regulation and to bring innovative strategies and methods for the treatment and management of LUT dysfunction.

### Funding

This research was funded by the Guangxi Science and Technology Base & Talents Fund (GUIKE AD22035948). The National Natural Science Foundation of China (No. 31970946) supported the APC of this article. The funders had no role in study design, data collection and analysis, decision to publish, or preparation of the manuscript.

## Grant Disclosures

The following grant information was disclosed by the authors:
Guangxi Science and Technology Base & Talents Fund: GUIKE AD22035948.

## Competing Interests

The authors declare there are no competing interests.

## Author Contributions

- Shutong Pang conceived and designed the experiments, performed the experiments, analyzed the data, prepared figures and/or tables, and approved the final draft.
- Junan Yan conceived and designed the experiments, authored or reviewed drafts of the article, and approved the final draft.

## Data Availability

   This is a literature review.

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
