# Peer review of "Research and progress on the mechanism of lower urinary tract neuromodulation: a literature review"

_PeerJ, doi:10.7717/peerj.17870_

## Round 0.1 · original submission · Major Revisions

Thank you for submitting your manuscript to our journal. We have thoroughly reviewed your work and value the contributions it makes to the field. While your submission demonstrates significant promise, the reviewers have identified several important concerns that need to be addressed. We encourage you to carefully consider and respond to each of these points, offering clear and detailed justifications for any revisions you make.

Reviewer 1 ·

Basic reporting

The manuscript by Pang and Yan addresses the mechanisms regulating lower urinary tract (LUT) function. This is a review aiming to provide the readers with knowledge about the neural pathways involved in control of LUT function. It also reports about methods commonly used to evaluate LUT function. Finally it also includes information about more recent approaches including the use of AI, machine-learning and data mining, which could prove to be useful to better understand urinary impairment and to make a more well-informed choice when choosing/advising on therapeutic approaches.
I read this manuscript attentively and, while it is well-written and the subject timely, the initial sections do not add a lot to the papers by Schneider et al (doi: 10.1002/nau.23455), Sartori et al (doi: 10.1016/j.euf.2019.12.004), Fowler et al (doi: 10.1038/nrn2401) and other studies. This is more evident as figure 2 is very similar to one found in Fowler et al. I suggest the authors revise the initial sections, making them more succinct and easier to understand. Also, neuromodulation is a therapeutic approach for many LUT dysfunctions. The authors use it as a synonym for neural regulation. This needs to be clarified.
The title of the section “Research methods for LUT modulation mechanisms” should changed to indicate that it refers to neural regulation. Indeed, there are non-neuronal modulation mechanisms, such as cross-organ sensitization for example, also interfere with LUT control.
In the section “Methods for LUT function evaluation”, I suggest a reorganization, indicating in the initial paragraphs which are the non-invasive and invasive methods being discussed.
The section about AI focused on human data (while the previsou sections describe animal research). It is a bit too general and vague. I was expecting more from this section. Nevertheless, as is, this manuscript is composed by two sections: the first on animal data, the AI part on potential clinical use.
Small points:
- There is no reference to figure 1.
- The source of figures or how they were generated should be indicated.

Experimental design

No comment.

Validity of the findings

No comment.

Additional comments

No comment.

Reviewer 2 ·

Basic reporting

Thank you to the authors for this important paper on the lower urinary tract and neuromodulation. Research and progress on the mechanism of lower urinary tract neuromodulation: a literature review.
This paper is well-written and illustrated. The implications for future work are important. One suggestion. Further comment on the promise and drawbacks of using surface electrodes, TENS for neuromodulation, would help; the tibial nerve.

Experimental design

well conducted review

Validity of the findings

justified

Additional comments

none

---

## Round 0.2 · accepted · Accept

Dear Authors,

After careful consideration and review of the revisions submitted, your manuscript has met the necessary standards for publication. We commend you for your diligent efforts in addressing the reviewers' comments and enhancing the quality of your work.

Reviewer 1 ·

Basic reporting

Overall, the manuscript has been improved.

Experimental design

Overall, the manuscript has been improved.

Validity of the findings

Overall, the manuscript has been improved.

Additional comments

Overall, the manuscript has been improved.